# Preclinical Testing Techniques: Paving the Way for New Oncology Screening Approaches

**DOI:** 10.3390/cancers15184466

**Published:** 2023-09-07

**Authors:** Antonia van Rijt, Evan Stefanek, Karolina Valente

**Affiliations:** 1Biomedical Engineering Program, University of Victoria, Victoria, BC V8P 5C2, Canada; antoniavanrijt@uvic.ca; 2VoxCell BioInnovation Inc., Victoria, BC V8T 5L2, Canada; estefanek@voxcellbio.com

**Keywords:** preclinical, oncology, 3D culture, organ-on-a-chip, spheroid, organoid, 3D bioprinting, drug screening

## Abstract

**Simple Summary:**

Traditional preclinical testing, including 2D cell culture and animal models, often fails to accurately predict drug efficacy in humans, especially for oncology drugs, where drug candidates that enter clinical trials have very high failure rates. Advancements in biology and tissue engineering techniques allow researchers to evaluate drug candidates before human trials using 3D cell culture models that more closely resemble human tissues than 2D culture methods. These techniques can better mimic the patterns of drug diffusion, cell–cell signalling, and the presence of vasculature in tumours in vivo. Furthermore, the FDA Modernization Act 2.0 promotes the use of higher complexity in vitro models such as 3D cell cultures. By offering more accurate representations of human tissue, 3D culture platforms have the potential to enhance preclinical drug development and lead to safer and more effective cancer treatments.

**Abstract:**

Prior to clinical trials, preclinical testing of oncology drug candidates is performed by evaluating drug candidates with in vitro and in vivo platforms. For in vivo testing, animal models are used to evaluate the toxicity and efficacy of drug candidates. However, animal models often display poor translational results as many drugs that pass preclinical testing fail when tested with humans, with oncology drugs exhibiting especially poor acceptance rates. The FDA Modernization Act 2.0 promotes alternative preclinical testing techniques, presenting the opportunity to use higher complexity in vitro models as an alternative to in vivo testing, including three-dimensional (3D) cell culture models. Three-dimensional tissue cultures address many of the shortcomings of 2D cultures by more closely replicating the tumour microenvironment through a combination of physiologically relevant drug diffusion, paracrine signalling, cellular phenotype, and vascularization that can better mimic native human tissue. This review will discuss the common forms of 3D cell culture, including cell spheroids, organoids, organs-on-a-chip, and 3D bioprinted tissues. Their advantages and limitations will be presented, aiming to discuss the use of these 3D models to accurately represent human tissue and as an alternative to animal testing. The use of 3D culture platforms for preclinical drug development is expected to accelerate as these platforms continue to improve in complexity, reliability, and translational predictivity.

## 1. Introduction

Preclinical testing is a cornerstone of the drug development process and is responsible for preliminary studies of numerous compounds with the ultimate goal of leading to a safe and effective drug. The drug development process outlined by the FDA includes a series of stages to gain approval for market placement [1]. During the preclinical stage, the screening of multiple compounds is performed with in vitro and in vivo platforms, including animal testing. Two-dimensional (2D) cell cultures are widely used in the preclinical stage to study the efficacy of cancer drugs, where cells are methodically exposed to therapeutic compounds and their responses are quantified. Through studying which concentrations of drugs are effective on various types of cells, the response of the same cell types can be estimated in vivo. Additionally, drugs can block specific signalling pathways in target cells, which can then be lysed and analyzed through biochemical or multi-omic approaches [2]. However, 2D cell cultures (cell monolayers) used in preclinical testing lack the diverse cell populations and structure of human tissue, including the extracellular matrix (ECM) and accessory solutes. In addition, 2D cultures deliver a constant concentration of drug over a uniform cell monolayer, which is uncharacteristic of the dynamics in which a drug will diffuse through a tissue or tumour [3].

Animal models are a prevalent tool in drug research and regulation, and they play a significant role in the preclinical stage. Through their integration in the pharmaceutical development and approval processes, animal models are utilized for their ability to mimic the response of the human body to prospective drugs [4]. Despite this, 90% of prospective drugs that pass the preclinical research stage fail in clinical trials, representing a significant inefficiency in the drug development process [5]. In addition, animal model results often have weak translatability to their associated clinical trials, and there is less than 8% correlation between in vivo data and clinical trial results [6]. Paired with the high failure rate of prospective drugs during preclinical testing, this reveals a stark need for preclinical testing techniques that more closely mimic human responses. This is especially true for anti-cancer drug development, where new therapies in the clinic have a failure rate of 95% [7].

Following the preclinical stage, clinical trials are performed in three phases of increasing patient volume: Phase 1 gauges the safety of the product, Phase 2 investigates its effectiveness, and Phase 3 studies the overall drug performance using a larger and more diverse patient population. If the drug passes clinical testing, it proceeds to the new drug application (NDA) review stage, and approved drugs are monitored once they are on the market in the post-marketing stage [8]. The drug development timeline is a time- and resource-consuming task, with the entire process leading to a single approved drug costing an excess of USD 1 billion over an average of 12 years [9].

In 2022, the United States Congress passed the FDA Modernization Act 2.0 to authorize alternative preclinical testing techniques as an exemption from animal testing [10]. In conjunction with recent developments in in vitro techniques, this could promote continuous developments and improvements in alternative preclinical models. Three-dimensional (3D) in vitro cell culture models offer significant benefits over 2D cell cultures in their mimicry of human physiology and phenotypic features; however, the benefits of many 3D models need to be further validated before they can be widely adopted into preclinical testing [11]. A variety of 3D in vitro methods can be used to validate drugs, including cell spheroids and organoids, organs-on-a-chip (OoCs), and 3D bioprinting techniques.

Cell spheroids are 3D aggregates of cells that better mimic the concentration gradient of a drug through a tissue when compared to 2D cultures [12]. Similar to spheroids, organoids are 3D structures of cells; however, they self-assemble using a hydrogel scaffold to form an extracellular environment and can have a variety of cell types with differentiated functions, allowing the organoid to have physiological properties closer to the intended organ [13]. While the lack of inherent vasculature in spheroids and organoids can allow for the modelling of hypoxia and the necrotic core of avascular tumours, these models fail to mimic truly vascularized tumours [14,15]. In addition, the lack of vasculature also inhibits nutrients from reaching the core of the structure, which can result in poor cell viability as their size increases and an inability to properly model drug delivery systems. Furthermore, the nutrient and oxygen gradients can influence cell growth, migration, and morphology [16].

Organs-on-a-chip (OoCs) are miniaturized microfluidic chips containing microscale tissues that more closely mimic natural tissue behaviour [17,18]. OoCs are advantageous in replicating the mechanical processes of organs during typical physiological conditions and can model cellular interfaces between different organ compartments [19,20]. They can also be extended to multi-organ models to study interactions between tissues from different organs, providing more of a systemic insight during the drug development process [21]. However, while OoCs have superior tissue mimicry compared to 2D cultures, the complexity of in vivo tissues is still much greater [22].

Three-dimensional bioprinting employs a variety of additive technologies combined with bioinks to produce 3D cellular structures [23]. Bioinks can be cell-laden or acellular, creating a scaffold for cells. Three-dimensional bioprinting allows for the fine tuning of the model’s 3D structure and a variety of bioinks can be used to closely replicate the mechanical structure of native human tissue, mimicking the ECM [24,25,26]. Common methods for 3D bioprinting include extrusion printing, digital light processing (DLP) printing, and two-photon polymerization (2PP). During extrusion printing, bioink is extruded through a nozzle and deposited layer by layer on a build plate to create a 3D structure [27]. DLP printing uses light to selectively crosslink bioink from a liquid vat to produce layers of the 3D structure [28]. 2PP utilizes the energy from the absorption of two photons using a femtosecond laser to precisely crosslink material in the focal point of the laser in three-dimensional space, or the voxel [29]. The resolution limitations of 3D bioprinting make it difficult to print vascularized structures to study the uptake of drugs through the circulatory system; while the average capillary diameter is 5 µm, bioprinting using hydrogels can yield resolutions of 100 µm through extrusion printing, 25 µm using DLP, and 100 nm using 2PP [30,31,32,33,34,35].

The advantages and limitations of in vitro preclinical testing platforms will be discussed with the ultimate goal of highlighting the most promising techniques for wider adoption with the support of the FDA Modernization Act 2.0.

## 2. Spheroids

### 2.1. Limitations of 2D Cell Culture

Two-dimensional cell cultures are the most widely used method for toxicological screening and determining drug candidate efficacy during preclinical testing. Two-dimensional cell culture consists of either using primary cells or immortalized cell lines and growing them in an artificial environment. Primary cells are derived from living tissue which are then cultured in vitro and eventually become senescent, while immortalized cell lines can be cultured indefinitely. Primary cells may be more physiologically relevant due to phenotypic variation in patient tissues; however, immortalized cell lines provide a consistent genetic profile and are easy to obtain [36]. Typically, cells must be attached to a solid substrate, but certain cultures can be grown in a suspension of the culture medium. Cultures that are anchored to a solid substrate form a monolayer culture where the substrate surface, such as a Petri dish, becomes covered by a single-cell-thick layer of cells as they attach and proliferate [3,37]. Two-dimensional cell cultures are easy to grow and proliferate quickly, and so they present a convenient way to gather preliminary data on drug behaviour [38]. However, 2D cultures do not fully depict human tissue due to the lack of 3D properties, and so they do not predict clinical physiological responses to compounds with accuracy. Two-dimensional monocultures do not mimic the diverse multicellular environment with multiple cell types and functions found in real tissues. Furthermore, 2D cultures struggle to provide insight on the systemic effects of drugs; 2D cultures can show how effectively a drug can kill cancer cells but cannot reliably predict complex phenomena such as organ toxicity. In addition, in 2D cultures, a constant concentration of the drug is deposited across the entire cell monolayer, whereas in vivo tissues will induce a concentration gradient from where the drug is delivered. As a result, drugs can be more effective on cancer cells in 2D, and so dosages based on 2D cultures may not be effective in 3D. Furthermore, native human tissue has additional constituents that exist alongside cells to regulate structure and function, such as the ECM, which gives mechanical structure to the tissue while also supplementing growth factors and other bioactive molecules to regulate cell characteristics [39]. Two-dimensional cultures also limit paracrine cellular interactions since signalling between adjacent cells is limited to two dimensions. The cell phenotype can also be altered when cultured in 2D, which can be partially mitigated by co-culturing in 2D with multiple cell types [40]. However, tumour tissue is composed of cancer cells along with healthy stromal cells, such as endothelial cells, stem cells, and fibroblasts, which biochemically support the tumour microenvironment and are involved in paracrine signalling [41,42]. Two-dimensional co-cultures still cannot fully recreate the tumour microenvironment due to their limited contact with adjacent cells and lack of tumour–stroma interactions.

### 2.2. Spheroids for Preclinical Drug Development

Cell spheroids were created to increase the predictivity of cell monolayers, taking preclinical testing to a further degree of complexity. Cell spheroids are 3D aggregates of cells typically formed by seeding cells onto a non-adherent substrate or hanging drop, allowing the cells to assemble into a 3D structure rather than forming a 2D monolayer. Various methods for establishing a spheroid culture are visually depicted in Figure 1. Spheroids are especially useful at modelling cancers, which have complex invasive behaviours, feedback mechanisms, and tumour–stromal cell interactions [43]. Spheroids form due to cell–cell adhesion from their integrin proteins and ECM proteins. Cells initially aggregate due to the binding of cell-surface integrin and leads to the upregulation of cadherin expression. This causes an accumulation of cadherin on the cell membrane, and the resulting cadherin–cadherin binding between cells strengthens the connections of adjacent cells to form the spheroid [44]. Integrins also play a role in the activation of focal adhesion kinase (FAK), a tyrosine kinase that participates in cell adhesion, migration, and growth. FAK influences cellular structure through the actin filaments in the cytoskeleton, which is integral for spheroid formation. In relation to cancer modelling, FAK overexpression is also associated with invasive tumour phenotypes, and so spheroids help to give a more faithful representation of tumour behaviour [45]. Spheroids also boast greater longevity than 2D cultures since they can be cultured for up to four weeks, whereas 2D cultures typically reach confluency within a week. Spheroids could thus be a better candidate for lasting studies to determine the long-term effects of drugs on surviving cells [38]. A summary of methods for formulating spheroids with their respective advantages and limitations can be found in Table 1. In summary, spheroids most closely capture tissue behaviour when they possess three key characteristics: a constitution of different cell types capable of cell–cell interactions, an ECM for mechanical stability and regulating cell function, and nutrient media with the required nutrients for tissue differentiation and maturation [43]. A summary of methods for establishing spheroid cultures can be found in Table 1.

### 2.3. Spheroids for Drug Efficacy

Cell spheroids’ mimicry of the tumour microenvironment and metastatic tumour behaviour can be harnessed in diverse ways to study anti-cancer drugs. Cancer spheroids have been used to study the effects of anti-cancer drug cisplatin on different cancer types [58]. The spheroids were formed using 96-well spheroid microplates and centrifugation, as well as different cell types including HeLa, A459, 293T, SH-SY5Y, and U-2OS. These spheroids were exposed to cisplatin, and ATP generation was monitored in real-time for 7 days, allowing for more long-term testing than that which is available for 2D cultures, which become contact-inhibited once confluent, resulting in behavioural changes from cell cycle arrest. In particular, the tumour–stroma interactions of the tumour microenvironment have been modeled using spheroids across various cancer types. Pancreatic stellate cells, a stromal cell type, have been cultured using spheroids with pancreatic cancer cells to investigate how the microenvironment influences the progression of pancreatic ductal adenocarcinoma. The co-cultured model exhibited resistance to gemcitabine and heightened migration when compared to the cancer cells cultured alone, emulating chemo-resistant, invasive, and metastatic phenotypes [59]. Spheroids have also been used to model metastatic pathways between different tissues for high-throughput drug testing. Co-culture spheroid hydrogels were previously used to produce a heterotypic model for prostate-to-bone cancer metastasis [60]. Despite breakthroughs in therapeutics, prostate cancer remains the second most frequently diagnosed cancer and the sixth leading cause of cancer death globally for men [61], and prognosis worsens for metastatic incidences of prostate cancer, which typically presents itself by the cancer spreading to the bone. Models for prostate-to-bone cancer metastasis were created by spotting in-air 3D droplets of prostate cancer cells co-cultured with osteoblasts onto a superhydrophobic surface. The spheroidal 3D microgels were composed of methacrylated hyaluronic acid (HA-MA) and methacrylated gelatin (GelMA). The co-cultured cells and physiological conditions were replicated using PC-3 cells initiated from the bone metastasis of a stage IV prostatic adenocarcinoma and human osteoblasts, which then exhibited mineralization through calcium deposits on the spheroid surface. The heterotypic 3D microgels demonstrated higher resistance to platin chemotherapeutics than comparable single or co-culture spheroids without the heterotypic prostate–bone interface [60].

### 2.4. Spheroids for Drug Toxicity

Previously, murine NIH3T3 fibroblasts were used as stromal cells in hanging-drop spheroid co-culture with human ovarian adenocarcinoma cells and pancreatic epithelioid carcinoma cells, respectively. The fibroblasts were engineered with a reporter gene whose signal could be monitored over time in media to determine the cytotoxic effects on the stromal cells compared to the cancer cells. This helped to identify a therapeutic window and provided opportunities to distinguish between broadly cytotoxic compounds and those that target cancer cells more selectively [62]. Co-cultured tumour spheroids have been used to study the efficacy of blocking cell membrane receptors NKG2A and MICA/B in therapies for colorectal cancer (CRC) [63]. NKG2A and NKG2D are receptors on natural killer (NK) immune cells, where NKG2A inhibits and NKG2D activates the NK function. These NK receptors have promising therapeutic potential for cancers such as CRC that are resistant to immunomodulation. The cellular infiltration of NK cells was enhanced by targeting NKG2D and its ligands MICA and MICB, resulting in a more-efficient destruction of CRC spheroids and demonstrating their potential for anti-cancer therapies.

## 3. Organoids

Like spheroids, organoids are 3D cell culture structures grown in vitro. However, organoids and spheroids have some key differences that give them strengths and limitations for various applications. Cell spheroids are aggregated together through weak forces such as protein interactions and adhesion molecules, and while they can be co-cultured to better recapitulate the tumour phenotype, they are relatively simple and do not possess the diverse differentiation of living tissues. Organoids, on the other hand, are more complex structures with differentiation and organization to mimic the function as well as the structure of a given organ [64,65]. Furthermore, while spheroids can be formed with discrete cell types, some types of organoids can differentiate into distinct populations of functional cells. However, while organoids offer greater complexity and functionally replicate target organs, they are more difficult to produce than spheroids, since spheroids can be generated easily and in large quantities due to their simplicity [66].

Organoids are usually cultured in a dome or layer of Matrigel, a soluble basement membrane extract from murine Engelbreth-Holm-Swarm sarcoma. Matrigel is composed of laminins, collagen IV, entactin, heparin sulfate proteoglycan perlecan, and a wide variety of growth factors [67]. The concentrations of individual components can vary significantly between batches, especially when considering the growth factors. As a 3D matrix for organoid culture, Matrigel provides an environment that mimics the basement membrane and promotes the growth and differentiation of epithelial cells; however, many cancers such as carcinomas invade the adjacent connective tissue and experience an ECM primarily consisting of collagen I fibers, that is very different to the composition of Matrigel. When culturing organoids derived from invasive carcinomas, researchers should consider the implications that this difference in matrix may have on the results and explore alternative options when feasible.

Organoids are prime candidates for investigations regarding specific aspects of organ function, while spheroids are better suited for high-throughput testing. Due to their advantages in modelling specific functions of human tissue, organoids have been harnessed to study intricate mechanisms of cancer, such as patient-specific drug resistance, genomics, and phenotype.

### Organoids for Preclinical Drug Development

Organoids derived from primary cancer cells are beneficial for studying variation in how cancers are presented across different patients and offer significant future possibilities for personalized medicine. In a previous study, organoids were derived from human lung, kidney, and gastric cancer cells from patients and cultured in Matrigel using a pair of synchronized modules to deliver droplets into a 96-well plate [68]. The first module, M1, is a microfluidic mixer that manipulates Matrigel and cools it to workable low temperatures. M1 forms droplets by combining Matrigel and cell solution with an immiscible oil, which form cell-laden Matrigel droplets in the cooled environment. The droplets are then heated in the same segment of tubing to transition to a gelled state. The second module, M2, then automatically dispenses the 3D organoid droplets in a 96-well plate. Using this automated system, the droplets developed into organotypic constructs larger than 400 μm within 5–7 days, rather than the 4–6 weeks traditionally required for growing organoids of this size, and were highly reproducible. The organoids conserved 97% of gene mutations from the primary tumour and were 80% accurate in recapitulating drug sensitivity variations between patients. As such, this automated organoid model shows promise for personalized medicine through rapid drug screening for cancer patients. Preclinical opportunities for organoids in cancer screening were also seen in a previous ovarian cancer (OC) model where tumour samples from epithelial OC patients were dissociated and cultured as organoids [69]. The organoids themselves recapitulated original tumour phenotypes, which were determined through immunohistological and immunofluorescence analyses targeting gene markers and the mutation of the tumour-suppressor protein p53, which is characteristic of the high-grade serous OCs investigated in this model. Furthermore, through DNA sequencing, the organoids reproduced the genetic structure of the primary tumours. Most importantly, the organoids exhibited sensitivities to clinical chemotherapy drugs that were congruent with individual patient responses. In conjunction, this OC model establishes that organoids can closely recapitulate primary tumour genomics and phenotype and can realistically model patient-specific chemotherapy responses when compared to clinical data. A cancer-screening organoid model was also studied for human gastric cancer [70]. In this model, organoids were sourced from gastric cancer patients with resected gastric cancer tumours and were subjected to standard-of-care chemotherapy. Organoids that showed resistance to the chemotherapeutics were sourced from patients with poor chemotherapy responses, indicating that the organoids were suitable for predicting patient drug sensitivities. Furthermore, RNA sequencing confirmed that the organoids closely resembled the primary tumour tissue.

## 4. Organ-on-a-Chip

OoCs are microscale systems that consolidate advancements in tissue engineering and microfluidic chips to mimic specific physiological tissue environments. Microfluidic chips use microscale channels that are configured in specific ways to manipulate small volumes of fluid, from picolitres to milliliters [71,72]. By integrating hydrogels mimicking miniature tissue models inside microfluidic chips, OoCs can circulate blood or nutrients to the tissue with a high degree of control. Using leading microfabrication technologies, OoCs can maintain highly controlled cell microenvironments and more closely recapitulate tumour phenotypes than 2D cell models [73]. Due to the decreased reagent consumption and increasingly simple fabrication methods associated with microfluidic technology, OoCs can reduce overall costs in preclinical testing while also better predicting physiological responses [74].

### 4.1. Single-Organ and Multi-Organ Chips

Single-organ models can be used to study the specific organ responses to a particular compound; additionally, the adaptable nature of microfluidic chips allows for multiple organ compartments to be integrated in one multi-organ system for monitoring interactions between different tissues in phenomena such as cancer metastasis or paracrine signalling [75]. Tissue models used in OoCs can include lab-grown organoids or primary samples obtained from patients. Lab-grown organoids are easier to grow in large quantities and have a relatively consistent genotypic profile, and so they can be beneficial in high-throughput drug screening. However, while they are complex structures, lab-grown organoids may not reflect the complete heterogeneity of human tumours, which can affect their responses to drugs. This is due to the importance of supporting cell types, immune cells, and the stroma in the tumour microenvironment. Furthermore, stromal cells and immune cells utilize signalling pathways to influence inflammation and tumour progression as well as ECM deposition [76,77]. Conversely, the benefits of primary samples include the fact that they retain the characteristics of the patient’s tumour, which allows for personalized medicine opportunities and reflects inter-patient differences in the drug response between patients with the same type of cancer [78]. However, primary samples are difficult to obtain in high quantities and have greater variation, making it difficult to interpret test results for high-throughput testing. In summary, cell source selection is dependent on the goals and purpose of the project.

### 4.2. OoC Architecture

Device architecture is another point of consideration in OoC design. Typically, OoCs can be categorized into two main construction styles that denote their general function and purpose. The first type is solid organ chips, which are comprised of 3D tissue masses that are positioned in the microfluidic chip such that they can interact with one another and the culture medium in specific ways. Solid organ chips can be used to model parenchymal or mesenchymal tissues for studying general tissue responses and properties [79]. The second type is barrier tissue chips, which have a structure that allows for the formation of a cellular barrier separating discrete fluid paths. This models the function of living barriers between endothelial and epithelial tissues and allows for the study of molecule transport or different responses between compartments [80]. Photolithography has long been used for the precise microfabrication of silicon, which offers opportunities for the integration of electronic components in OoCs; however, it is brittle and has poor optical properties [81]. Other commonly used materials for the fabrication of OoCs include polydimethylsiloxane (PDMS), glass, 3D printing resins, and thermoplastics, such as polymethylmethacrylate (PMMA) and cyclic olefin copolymer (COC), each with their own benefits and drawbacks as introduced in Table 2. PDMS is high-resolution and has a relatively inexpensive and simple fabrication process using soft lithography, but is susceptible to absorbing and subsequently leaching various chemicals which can be detrimental to experiment reproducibility [82,83]. Glass OoCs are inert but are significantly more expensive to produce and have extensive manufacturing processes [84]. Three-dimensional printing resins offer benefits for iterative testing due to their rapid prototyping but have poor optical properties and little supporting literature for resin biocompatibility [85,86]. Thermoplastics are suitable for injection molding in mass-produced OoC systems, but can have sealing difficulties since typical bonding techniques used in soft lithography do not translate well to PMMA chips [87,88,89]. As there are many benefits and drawbacks surrounding OoC fabrication, each design should be assessed to determine the most suitable method.

### 4.3. OoC for Preclinical Drug Development

OoCs hold promise for cancer drug screening by offering physiological relevance while retaining experimental controllability and reproducibility [18]. Cardiotoxicity is a cause of concern for anti-cancer drugs since cardiac safety is not always recapitulated in current preclinical animal models. The human ether-a-go-go-related gene (hERG) encodes a subunit of a potassium channel and plays a significant role in cardiac repolarization. Blocking hERG leads to long QT syndrome and associated fatal arrhythmias The blockage of hERG is the most common cause of cardiotoxicity, and as a result, evaluating the effect of drugs on hERG can indicate prospective cardiotoxicity levels [91]. An integrated OoC was previously created to evaluate anti-tumour drug efficacy and cardiac safety for linsitinib as a treatment for Ewing Sarcoma (ES), which demonstrated promise for treating ES in preclinical xenograft models but had a high incidence of patients with relapsed or refractory ES in clinical testing [92]. The polysulfone-based OoC with cultured bone ES tumour tissues for recreating the ES microenvironment and heart muscle tissues was circulated with linsitinib and its response to the drug was compared with clinical results. The results from this integrated setting exhibited minimized tumour reduction and less cardiotoxicity than the xenograft models that predicted significant decreases in tumour and cardiac function. In summary, the OoC system had more congruent results with clinical trial data than the current xenograft model, and had opportunities to be extended to other drug and tissue systems. OoC can also be harnessed to mimic vascularized structures due to their ability to finely manipulate very small quantities of fluid. OoCs have previously been used to produce vascularized micro-organs and micro-tissues for studying the effects of anti-cancer drugs using self-assembled vascularized tissues [93]. In this study, a device was fabricated using a PDMS microfluidic layer with three connected chambers and a transparent polymer membrane bonded to a bottomless 96-well plate, and self-assembled vascularization was achieved through the cell-laden gel matrix by subjecting the chambers to gravimetric flow for one week. Additionally, healthy vascularized micro-organs (VMOs) were fabricated using normal human lung fibroblasts and endothelial progenitor cell lines, with a colorectal cancer cell line added to the vascularized micro-tumours (VMTs). Vascular permeability was quantified using mathematical models, and the tissues were perfused with chemotherapy drugs fluorouracil, vincristine, and sorafenib to monitor their influence on the vasculature and surrounding structures. The microfluidic chip was able to successfully form self-assembled vascularized models and confirmed that vincristine is a vascular disrupter while fluorouracil is not. Furthermore, the platform boasted high reproducibility, indicating its potential use for high-throughput testing. A similar three-chamber OoC was used to create colorectal cancer VMTs where the structures better recapitulated the gene expression and chemotherapy responses of parallel xenograft tests when compared with monocultures and 3D spheroids [94]. Additionally, the VMT vasculature exhibited leakiness that was associated with in vivo tumours and better mimicked the heterogeneous tumour microenvironment by recreating tumour–stroma interactions that were correlated with native human tumours.

### 4.4. Vascularization Capabilities of OoCs

Vascularized OoCs for phenotypic screening have also been fabricated using injection molding for high-throughput testing. A microfluidic system was previously designed using sequentially patterned hydrogels in a microfluidic body to induce the formation of perfusable vascular networks [95]. The platform setup, dubbed the MicroVascular Injection-Molded Plastic Array 3D Culture (MV-IMPACT), is an injection-molded device, allowing it to be produced in high volumes with little variation. The assembly of vascular networks in a central section of fibrin gel seeded with endothelial cells was facilitated by lung fibroblasts in fibrin gel on each side, and human colorectal adenocarcinoma and hepatocellular carcinoma were incorporated to recapitulate heterogeneous tumour–stroma interactions of the tumour microenvironment. The MV-IMPACT was used to validate the efficacy of DAPT, a drug associated with the formation of thicker and more numerous angiogenic vessels through Notch signalling pathway inhibition. Groups treated with DAPT exhibited increased vessel thickness and branching, which was congruent with similar experiments using in vivo techniques. By utilizing injection molding and a compact design, the platform showed promise for high-throughput testing with decreased reagent consumption, increasing in vitro test efficiency and accessibility.

## 5. Three-Dimensional Bioprinting

Three-dimensional printing is an additive fabrication technique that utilizes sequentially produced layers that are combined to produce a final 3D structure [96]. Using thermoplastics, photosensitive resins, or metal, 3D printing can produce intricate rapid prototypes for iterative testing and can also be used in the final manufacturing stages [97,98]. Three-dimensional bioprinting combines these principles with biological materials, cells, and supporting biomolecules to produce detailed tissue-like 3D structures that can closely mimic real tissue physiology. Three-dimensional bioprinting is beneficial for tissue modelling as it can be used to produce vascularized structures and their biochemical and mechanical properties can be finely tuned through bioink selection [99]. The overall biomimicry of tissue models is dependent on the specific method of 3D bioprinting and biomaterials used [100,101]. A summary of common 3D bioprinting methods can be found in Figure 2.

### 5.1. Extrusion 3D Bioprinting

Extrusion bioprinting dispenses bioink through a nozzle onto a substrate using a series of motors for precise movement of the printhead and building stage. Upon being dispensed by the nozzle, the printed material may be either ionically crosslinked or photo-crosslinked [102,103]. Furthermore, multi-printhead systems can be used to produce structures with a variety of materials for more heterogeneous structures or for sacrificial supports. Bioinks with shear-thinning properties can be beneficial for extrusion bioprinting as the sharp convergence of the nozzle causes these materials to decrease in viscosity, allowing the bioink to flow through the nozzle more easily and become more structurally stable again from their increased viscosity after deposition. This can also improve cell viability since unregulated shear force at the nozzle can damage cells and lead to cell death [104,105]. Extrusion bioprinting can be performed with very high cell densities, which is important for accurately recreating native tissues [106]. Resolutions of 100–600 µm can be achieved through extrusion bioprinting, and producing intricate structures such as vascular networks can be challenging [31,32].

### 5.2. Digital Light Processing (DLP) 3D Bioprinting

DLP is a type of light-based 3D bioprinting that utilizes specific wavelengths of light to selectively polymerize liquid bioink in a reservoir. Polymerization is achieved through the activation of a photoinitiator in the bioink. DLP polymerizes an entire layer of bioink at a time using a digital micromirror device (DMD) projector [107,108]. As such, bioinks are restricted to light-activated materials. Resolutions of 25–100 µm can be achieved through DLP bioprinting, producing more complex structures such as vascular networks with some success, especially with synthetic polymers [107]. DNA damage from UV exposure and cytotoxic species formed through the photoactivation process can both be detrimental to cell health, but photoinitiators with improved biocompatibility that are active in visible light spectra such as LAP have reduced the severity of this issue [109].

### 5.3. Two-Photon Polymerization (2PP) 3D Bioprinting

TPP 3D bioprinting is a direct writing method that utilizes a femtosecond laser to selectively polymerize bioink with a high degree of precision. TPP harnesses two-photon absorption theory, where the simultaneous absorption of two photons by a photosensitive molecule increases the energy state of the molecule, initiating the polymerization of the material at the focal point of the laser [33]. Since crosslinking only occurs at the voxel, light can pass through areas of the material without polymerizing it. The polymerization at the voxel can yield resolutions down to 100–400 nm, allowing for bioprinting detailed structures with high precision, including more intricate vascular patterns [110]. While various photoinitiators exhibit promising biocompatibility, some may negatively impact cell viability through the potential to produce volatile species during their activation mechanism [111,112]. Furthermore, the wavelengths of light used in 2PP need to be considered to avoid cell damage due to irradiation at harmful wavelengths, such as UV light leading to apoptosis [113].

### 5.4. Three-Dimensional Bioprinting Biomaterials

Three-dimensional bioprinted structures are also characterized by their bioink constituents. Bioink components can be typically categorized into either natural or synthetic biomaterials, which have overarching traits that are common to each respective group. Natural biomaterials have complex biochemical traits and can closely mimic the ECM. This leads to favourable biocompatibility and cell adhesion, which allows cells to proliferate, differentiate, and migrate similarly to native tissues. However, due to inherent inconsistencies between sources of natural substances, naturally derived biomaterials often have significant variance between batches, making it difficult to produce bioinks with highly reproducible characteristics. Furthermore, they often have weak mechanical properties, and so they may not be fit for mimicking more rigid tissues and are prone to degradation. Conversely, synthetic biomaterials have low variability and are highly reproducible, allowing for the fine tuning of mechanical properties and degradation through the modification of functional groups of the polymer backbone. However, synthetic biomaterials do not mimic the ECM as closely as natural materials and may produce cytotoxic by-products upon degradation that can be detrimental to biocompatibility [114]. A variety of commonly used natural and synthetic biomaterials can be found in Table 3 and Table 4, respectively. As such, bioinks that utilize both natural and synthetic biomaterials simultaneously can reap the benefits of both types while minimizing their limitations.

### 5.5. Three-Dimensional Bioprinting for Preclinical Drug Development

Due to its high degree of spatial control, 3D bioprinting shows significant promise for producing physiologically relevant preclinical models for a variety of cancer subtypes. An overview of recent studies using 3D bioprinting to create preclinical models for different cancer types can be seen in Figure 3. Breast cancer is among the most prevalent cancer types worldwide and has an abnormally high risk of death after diagnosis for many more years compared to other cancer types [160]. As such, breast cancer is the most commonly researched cancer type among experimental papers [161]. Laser-based 3D bioprinting techniques have been previously used to produce breast cancer spheroids with high spatial control. The printing system used laser direct writing (LDW) for patterning microstructures with cell-laden biomaterials where alginate loaded with human MDA-MB-231 cells was selectively ejected into microbeads and crosslinked using calcium ions to produce core-shelled breast cancer structures [28]. This model was especially useful for modelling the hypoxic and necrotic nature of tumour cores with a high degree of reproducibility, as excessively small microbeads would have insufficient hypoxia at their core, but excessively large microbeads would have an overly necrotic core which may not be characteristic of in vivo conditions due to angiogenesis and tumour vascularization. As such, this model shows promise for utilizing 3D bioprinting methods in high-throughput testing with high accuracy and precision.

### 5.6. Three-Dimensional Bioprinting for Modelling Metastasis

Furthermore, tissues with diverse structural and mechanical properties can be fabricated using 3D bioprinting. The most common metastatic site for breast cancer is the bone, and prognosis sharply worsens for patients once a metastatic state has been reached [164]. Traditional biomaterials can be used for modelling most soft tissues; however, a previous study incorporated nanocrystalline hydroxyapatite into GelMA bioink to study the interactive effects of cells in artificial bone microenvironments [162]. SLA printing was used to fabricate bone matrices laden with osteoblasts and bone marrow mesenchymal stem cells, to which breast cancer cells were later introduced. Upon being introduced to the bone matrices, the breast cancer cells had enhanced growth and increased vascular endothelial growth factor. Conversely, the stromal bone cells had decreased proliferation and reduced alkaline phosphatase activity. In conjunction, these results indicate that 3D bioprinted co-cultures can help to model the behaviours of complex metastatic cancer in post-metastatic stages.

### 5.7. Vascularization Capabilities of 3D Bioprinting

Three-dimensional bioprinting is attractive due to its possibilities for vascularization strategies. Compared to other 3D cell culturing techniques, 3D bioprinting offers the most promise for producing specific and intentional vascular networks, especially with higher-resolution printing methods. This can be used to study various vascular phenomena of cancer, such as metastasis, angiogenesis, and oncologic drug uptake. A previous study utilized DLP techniques to fabricate honeycomb vascular-like structures with a variety of channel widths using PEGDA and studied the circulation of HeLa cervical cancer cells versus noncancerous 10T1/2 cells [165]. While the noncancerous cells showed similar migration patterns regardless of channel width, the HeLa cells exhibited greater migration as the channel width decreased. This trend is congruent with the cellular arrest and distant site proliferation characteristics of the metastatic process and suggests that 3D bioprinted vascular networks can be valuable tools in modelling metastatic cancers.

The vascular capabilities of 3D bioprinting can also be harnessed to produce 3D tissue cultures for diseases, for which there are limited biomimetic models. Vascularized 3D bioprinted models have previously been studied for neuroblastoma (NB) by bioprinting multi-channel structures in HUVEC-lined GelMA and seeding NB spheroids at the core [166]. The vascularized outer structure was created by extruding GelMA into a support bath before being cured by UV, which were then seeded with HUVEC cells post-crosslinking. NB spheroids were cultured separately and were then manually added to the HUVEC-lined vascularized GelMA structures which were then dynamically cultured by rocking. The model exhibited an increase in NB aggressive behaviour through tumour cell migration and the integration of NB into the surrounding endothelial structures. Furthermore, by assessing the model’s gene expression profile, the mechanisms for NB therapeutic responses can be studied.

## 6. Conclusions and Future Outlook

Three-dimensional cell culture models mimic many aspects of the tumour microenvironment, which allows them to provide preclinical insights on drug efficacy and toxicity that are not possible to obtain with 2D cultures. Although existing 3D culture models lack the complex interplay between organ systems that occurs in animals, these in vitro models could be a viable alternative for much of the preclinical testing performed in animals, including studies of target-tissue efficacy, liver toxicity, or kidney toxicity. The continuous development and validation of 3D culture models is required for these platforms to become adopted in the drug development process on larger scales. Validating the clinical relevance of results from 3D culture models with results from clinical trials is particularly important to demonstrate the advantages of 3D culture models. Although this can be performed retroactively, the strongest evidence on the benefits of 3D culture models or any new preclinical testing model is provided when the platform is used to guide the development of an active therapeutic, and its clinical predictions are later confirmed in clinical trials. Three-dimensional culture models that are more standardized and thus relevant for use with drug candidates across a variety of indications will see quicker and wider adoption in the drug development process. Advantages of standardized 3D culture models are that they can be used across multiple pipelines within the same company, regulatory bodies are more likely to be familiar with interpreting their results, and much less time and resources are required to initially validate the model than with unique models.

Since animal models often fail to produce results that fully translate to clinical trials, 3D cultures also have a significant opportunity to provide better outcomes for patients by increasing the likelihood that innovative therapies succeed in clinical trials. Reducing the number of drug candidates that enter clinical trials to ultimately fail due to toxic side effects will also alleviate many of the adverse events that patients unfortunately experience in clinical trials. Discrepancies between animals and humans in metabolism, physiological differences, and disease states all contribute to the lack of correlation between animal and clinical tests. Additionally, drug sensitivities and metabolic processes can differ between animal and human models which can cause disparities between how animals and humans react to prospective drugs.

## Figures and Tables

**Figure 1 cancers-15-04466-f001:**
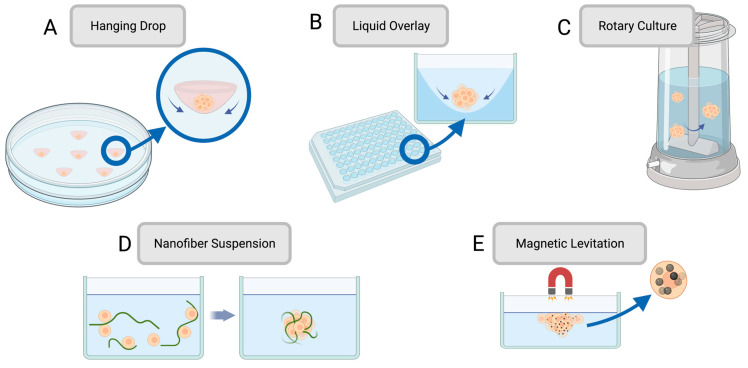
Common methods for producing spheroids for preclinical drug development such as (**A**) hanging drop, (**B**) liquid overlay, (**C**) rotary culture, (**D**) nanofiber suspension, and (**E**) magnetic levitation. Created using BioRender.com.

**Figure 2 cancers-15-04466-f002:**
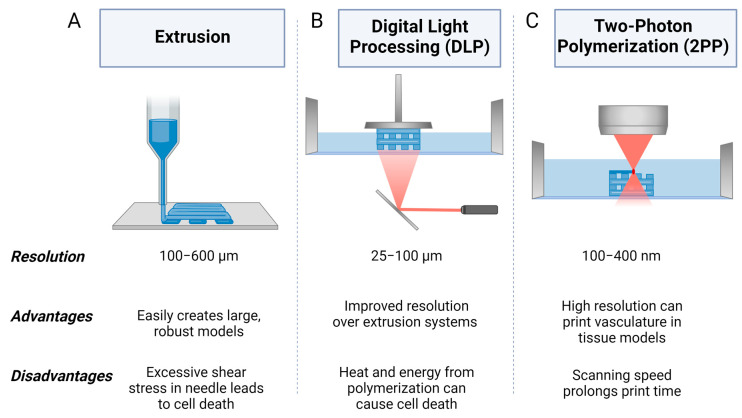
Schematic of common methods of 3D bioprinting including (**A**) extrusion, (**B**) digital light processing (DLP), and (**C**) two-photon polymerization (2PP). Created using BioRender.com.

**Figure 3 cancers-15-04466-f003:**
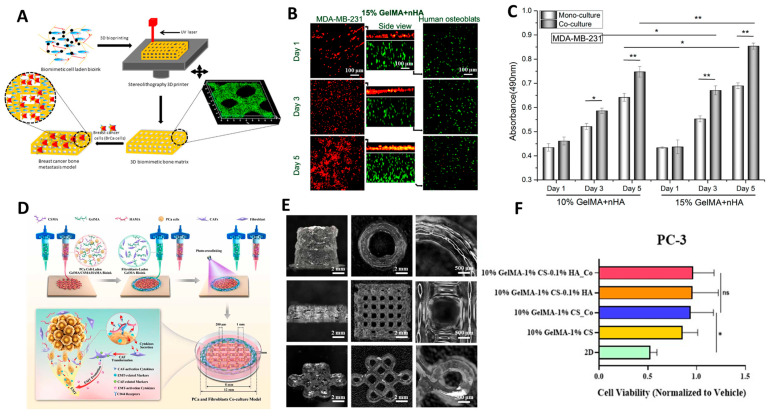
Recent 3D bioprinted cancer models for preclinical testing using (**A**–**C**) breast cancer and osteoblast co-culture [162] and (**D**–**F**) prostate cancer and fibroblast co-culture [163]. (**A**) Schematic diagram of light-activated 3D bioprinting process using cell-laden osteoblast scaffolding seeded with breast cancer cells to model breast cancer metastasis. (**B**) Confocal images of osteoblasts and breast cancer cells in co-culture after 1, 3, and 5 days. (**C**) Proliferation of breast cancer cells when cultured in mono-culture versus metastatic co-culture model with osteoblasts. (**D**) Schematic diagram of 3D bioprinting extrusion process for prostate and fibroblast co-culture model. (**E**) Extrusion printing of complex structures for prostate–fibroblast model. (**F**) DTX drug resistance of PC-3 (prostate cancer) cells in co-culture model. * indicates *p* < 0.05 and ** indicates *p* < 0.01. (**A**–**C**) Reprinted (adapted) with permission from [162]. Copyright 2016 American Chemical Society. (**D**,**E**) Reprinted under terms of the Creative Commons Attribution 4.0 License accessed on 24 August 2023 from https://creativecommons.org/licenses/by/4.0/ [163].

**Table 1 cancers-15-04466-t001:** Technical overviews of spheroid fabrication methods with respective advantages and disadvantages.

Spheroid Fabrication Method	Overview	Advantages	Drawbacks	References
Hanging drop	A drop of cell suspension is placed onto the inside of a cell culture plate lid, which is then inverted without disturbing the droplets held by surface tension. Over time, cells are concentrated and cluster into a spheroid at the bottom of the hanging droplet.	Simple Requires no specialized equipment Can be used with small cell suspension volumes	Laborious Low throughput High shear force Limited cell lines form spheroids through this method	[46,47]
Liquid overlay	Cell suspension is seeded onto a non-adherent surface with recesses that promote cell aggregation.	High-throughput Control over spheroid size	Some cell lines may need added ECM proteins to promote spheroid formation	[48,49]
Rotary cell culture	Cells are cultured in a container with an agitator that disrupts the cells’ ability to adhere to the substrate, forcing them to self-assemble into spheroids.	Simple High-throughput Large scale	Spheroid size variation Viability challenges due to mechanical damage	[44,50,51]
Nanofiber cell suspension	Adding polymer nanofibers to the cell suspension increases spheroid production due to cells interacting with the nanofibers.	Reduced cell death due to non-adherence Suitable for anchorage-dependent cells More time-efficient than other adaptations for anchorage-dependent cells	Polymer nanofibers may have unintended impacts on cell behaviour	[52,53,54]
Magnetic levitation	Magnetic particles are combined with cells, and a magnetic force is introduced. Negative magnetophoresis induces a weightless environment where cell aggregation is promoted.	Low-cost Allows for real-time imaging Minimizes additional forces on cells	Can lead to apoptosis	[55,56,57]

**Table 2 cancers-15-04466-t002:** Comparison of common materials for OoC including their suitable fabrication methods and the advantages and drawbacks for each.

Material	Fabrication Method	Advantages	Drawbacks	References
Polydimethylsiloxane (PDMS)	Soft Lithography	Optically clear Recapitulates high detail Easy fabrication Permeable to gasses Hydrophilic/hydrophobic capabilities Biocompatible	Absorption, retention, and release of small molecules Small molecule absorption and release can lead to hormone disruption, such as the absorption and release of estrogen associated with breast cancer studies Laborious for mass production	[82,83,90]
Polymethylmethacrylate (PMMA)	Injection Molding	Optically clear Minimal absorption Cost-effective for mass production	High stiffness Low fidelity in complex microstructures Low gas permeability Difficult to seal	[88,89]
Cyclic olefin copolymer (COC)	Injection Molding	Optically clear Minimal absorption Cost-effective for mass production	High stiffness Low fidelity in complex microstructures Low gas permeability Difficult to seal	[87]
Silicon	Photolithography	Compatible with electronic integration Versatile surface treatments Recapitulates high detail	Laborious and costly to produce Requires cleanroom facilities Poor optical transparency Brittle	[81]
Glass	Etching	Optically clear Inert Chemically resistant Biocompatible	Laborious and costly to produce Brittle	[84]
Resins	3D Printing	Low cost Rapid prototyping High-throughput	Poor optical properties Poor biocompatibility Low permeability Texturally rough Low fidelity in complex microstructures	[85,86]

**Table 3 cancers-15-04466-t003:** Summary of common natural biomaterials with properties and crosslinking mechanisms for use in bioinks.

Material	Overview	Properties for Bioinks	Crosslinking Mechanisms	References
Collagen	Triple helical protein for tissue scaffolding and tensile strength in tendon, cartilage, bone, and skin	Biodegradable Biocompatible Contributes to printability Bioactive properties	Covalent bonding of fibrils	[115,116,117]
Gelatin	Hydrogel from the hydrolysis of collagen, solid when cooled, and can be used to synthesize gelatin methacryloyl (GelMA)	Temperature-based gelation Printable Tunable mechanical properties	Gelation under cold temperatures	[118,119,120,121]
Gelatin Methacryloyl (GelMA)	Gelatin derivative with methacrylated functional groups; mechanically stable after photocrosslinking	Selective crosslinking Mimics the ECM Cell-binding sites Biocompatible Tunable	Photocrosslinked under UV light exposure	[122,123]
Fibrin	High viscosity; insoluble biopolymer that allows for paracrine signalling due to non-linear elasticity	Biocompatible Biodegradable Regenerative Nanofibrous structural properties Imitates both hard and soft tissues	Cleaved by thrombin which induces polymerization	[124,125,126]
Hyaluronic Acid	Bioresorbable material found in mammalian ECM; maintains a hydrated environment	High porosity allows for compound diffusion Must be combined with other biomaterials for bioink synthesis as it lacks mechanical stability and cell adhesion alone	Enzyme-crosslinking, Schiff base reaction, thiol-modified HA crosslinking, Diels–Alder click crosslinking, ionic crosslinking, and photo-crosslinking	[127,128,129,130,131,132]
Chitosan	Polysaccharide derived from chitin deacetylation with solubility at low pH levels	Nontoxic Bio-adhesive Suitable for soft tissues due to low mechanical strength	Chemical crosslinking with glutaraldehyde (amine groups) or citric acid (covalent)	[133,134,135]
Alginate	Polymer derived from brown algae, can form hydrogels that mimic the ECM, and can be crosslinked through its aldehyde groups	Biocompatible Low cost Low bioactivity Can degrade easily due to hydrolytic degradation	Ionically crosslinked with divalent cations	[136,137,138,139]
Decellularized ECM	Produced by removing cellular components from tissues via chemical or physical processes	Can retain tissue-specific behaviours post-decellularization May not require additional crosslinking	Glutaraldehyde; thermal gelation	[140,141,142,143]

**Table 4 cancers-15-04466-t004:** Summary of common synthetic biomaterials with relevant properties for bioink applications.

Material	Overview	Properties for Bioinks	References
Polylactic acid (PLA)	Semi-crystalline structure with high molecular weight, used in extrusion-based bioprinting	Useful for dental models Accurate surface properties Can be brittle	[144,145,146]
Poly(lactic-co-glycolic acid) (PLGA)	Synthesized through co-polymerization of both glycolic acid and lactic acid	Cell-compatible Can perform controlled drug release Properties can be tuned through glycolic to lactic ratio	[147,148,149,150]
Poly(ethylene glycol) diacrylate (PEGDA)	Long-chain photo-crosslinkable monomer that forms hydrogels	Photo-crosslinkability allows for use in light-based printing Hydrophilic for cell maintenance and encapsulation Modular though tunable functional groups	[151,152,153]
Poly e-caprolactone (PCL)	Semi-crystalline thermoplastic with high thermal stability and long degradation rate	Rubber-like flexibility in physiological conditions High permeability Useful for bone models due to degradation rate	[154,155,156]
Poly(propylene fumarate) (PPF)	Linear unsaturated polyester with fumaric acid backbone chains	High viscosity Light-responsive Useful in degradable materials as its ester bonds can be hydrolyzed, allowing for excretable products	[157,158,159]

## Data Availability

The data presented in this study are available in this article.

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
