# Peer review of "Preclinical Testing Techniques: Paving the Way for New Oncology Screening Approaches"

_cancers, 2023, doi:10.3390/cancers15184466_

Round 1

Reviewer 1 Report

The review by van Rijt et al. is a useful summary of 3D culture techniques. The context includes a timely nod to the new FDA law that encourages alternatives to animals for pre-clinical work.

The review is perhaps a little too strongly written in places. E.g., the "simple summary" states that 3D cultures "more closely resemble human tissues than traditional methods", which would be fine if the comparison were only to 2D culture, but "traditional" had previously been defined to include animal models. Similarly, line 59 claims that "animal model results are not translatable", which should probably be modified by stating that their translatability is weak rather than completely lacking.

The main problem with the manuscript is that the formatting of the tables is poor, with indistinct breaks between sections.

A minor problem is that the focus of the review is on human systems, but line 214 refers to a co-culture model with NIH-3T3 fibroblasts without noting that these are murine cells.

Another minor problem comes from the discussion of the positives and negatives of the materials in Table 2.  The note that comes with PDMS about absorption and release of small molecules should probably be strengthened by including a specific example like estrogen, which should be a major concern for the many groups trying to work with breast cancer. 

The section titles on line 400 "5.3. D Bioprinting" and on line 455 "5.4.3. D Bioprinting Biomaterials" seem to have both been confused from including "3D".

Line 46 "can be subjected to cells" and line 53 "a prolific tool" and line 203 "metastasis through the bone" are all odd usages.

Lines 72-75 seem to be making the argument that the new FDA Modernization Act will lead to "increasing accuracy of preclinical models".

Author Response

Dear Reviewer 1,

First of all, we would like to thank the reviewers for their comments and feedback. Our manuscript titled “Preclinical Testing techniques: Paving the Way to New Oncology Screening Approaches” has been significantly improved according to the feedback of the reviewers. We have tried to address most of the comments of the reviewers and the details can be found below. The reviewer questions can be found in bold, our answers can be found in italic, and the changes done in the manuscript can be found in blue.

The manuscript was updated, and the changes can be observed by the use of track changes. We hope that the changes created in the manuscript are satisfactory and we welcome any additional feedback.

Dr. Karolina Valente

Reviewer #1:

  • The review is perhaps a little too strongly written in places. E.g., the "simple summary" states that 3D cultures "more closely resemble human tissues than traditional methods", which would be fine if the comparison were only to 2D culture, but "traditional" had previously been defined to include animal models. Similarly, line 59 claims that "animal model results are not translatable", which should probably be modified by stating that their translatability is weak rather than completely lacking.

Thank you, these sentences have been modified in the manuscript.

“Advancements in biology and tissue engineering techniques allow researchers to evaluate drug candidates before human trials using 3D cell culture models that more closely resemble human tissues than 2D culture methods.

In addition, animal model results often have weak translatability to their associated clinical trials, and there is less than 8% correlation between in vivo data and clinical trials results [6].”

  • The main problem with the manuscript is that the formatting of the tables is poor, with indistinct breaks between sections.

This has been modified and the table has been updated. Light grey shading has been applied to alternating rows on each table to add distinction to different table sections.

  • A minor problem is that the focus of the review is on human systems, but line 214 refers to a co-culture model with NIH-3T3 fibroblasts without noting that these are murine cells.

Thank you. This has been corrected and updated.

“Previously, murine NIH3T3 fibroblasts were used as stromal cells in hanging-drop spheroid co-culture with human ovarian adenocarcinoma cells and pancreatic epithelioid carcinoma cells, respectively.”

  • Another minor problem comes from the discussion of the positives and negatives of the materials in Table 2.  The note that comes with PDMS about absorption and release of small molecules should probably be strengthened by including a specific example like estrogen, which should be a major concern for the many groups trying to work with breast cancer. 

Table 2 has been updated according to the feedback.

  • The section titles on line 400 "5.3. D Bioprinting" and on line 455 "5.4.3. D Bioprinting Biomaterials" seem to have both been confused from including "3D".

These section titles have been updated to the following to alleviate confusion:

  1. Three-Dimensional Bioprinting

5.4. Three-Dimensional Bioprinting Biomaterials

  • Line 46 "can be subjected to cells" and line 53 "a prolific tool" and line 203 "metastasis through the bone" are all odd usages.

Those lines have been updated accordingly.

“Additionally, cells can be influenced by drugs with mechanisms that block specific signaling pathways, which can then be lysed and analyzed through biochemical or multi-omic approaches [2].

Animal models are a prevalent tool in drug research and regulation, and they play a significant role in the preclinical stage.

Despite breakthroughs in therapeutics, prostate cancer remains the second most fre-quently diagnosed cancer and the sixth leading cause of cancer death globally for men [61], and prognosis worsens for metastatic incidences of prostate cancer, which typically presents itself by the cancer spreading to the bone.”

  • Lines 72-75 seem to be making the argument that the new FDA Modernization Act will lead to "increasing accuracy of preclinical models".

This sentence has been modified accordingly.

“In conjunction with recent developments in in vitro techniques, this could promote the continuous development and improvement of alternative preclinical models.”

Reviewer 2 Report

Review of the manuscript: Preclinical Testing Techniques: Paving the Way to New Oncology Screening Approaches

Authors present a review manuscript on Preclinical Testing Techniques: Paving the Way to New Oncology Screening Approaches. The author's review discusses common forms of 3D cell culture, including cell spheroids, organoids, organs-on-a-chip, and 3D bio-printed tissues. The authors summarized the advantages and limitations and discuss the use of 3D models as alternatives to animal testing shedding light on the urgency of getting new and stronger 3D models as preclinical testing techniques. This fact would prompt faster and better-fitting drug development that would lead to safer and more efficient therapies. 

In a wide range, the manuscript presents a standard organization, is clear and well written but, some amendments should/must be done to achieve a better outcome and to be considered for publication. Points raised will follow bellow:

The introduction, like all the sections of the presented manuscript, is clear and well-written, well-structured, and raises the most relevant points on the field of research, points that were mentioned initially in the manuscript abstract. Even if some issues should be addressed better. 

Small issues in text:

Simple summary and abstract are well-structured and written. The reading is clear and the main points are easily understandable. In the Abstract, line 26, please change the word “realistic” to another one. Realistic doesn’t seem to be the best term in the context of the sentence.

In the introduction part, at line 46 and 47, please take not that cells are subjected to drugs not the other way around. Studies are done to verify the cellular outcome of certain drugs. Please change the orientation of this sentence.  

At line 60, please re-insert reference 3. This appears as a superscript and not with [ ]. Please check and correct this with your Mendeley/Endnote bibliographic editor software. Synchronize all the references.

In the introduction at line 134, the authors mentioned the fact that 2D cultures do not mimic 3D culture environments or tissue ones. Undeniably, this is absolutely true. Still they should have been more precise on the line 134 “instead of …so they may not predict clinical….” Please state that “they not predict clinical…, cancel the word “may”. The entire scientific community knows and is aware that 2D model don’t mimic at al tissue environments due to many reasons. Please be more critical on the manuscript, if sustained by refs, authors scientific opinion is always welcome.

Methodologies unfolding spheroid formation are quite well described and illustrated in Figure 1. The figure is clear and schematically describes the different methodologies. It should be centered on the page having the text as a reference. Please do the same for Figure legends.  If this concerns to authors, please changed it, otherwise don’t consider this point. Table 1 is well-structured, has scientific references, and is easy to understand.

Like mentioned before, all sections are well written and designed. Authors made very good effort on the description of laboratory techniques used to perform preclinical tests on oncology broad field. Beyond a description of the current state-of-the-art of the field, I do believe that all the sections lack personal scientific criticism. Authors should explain why e.g. research is done using Matrigel for decades even if spheroids/organoids of different sources, in a regular situation would never be surrounded by a basement-membrane matrix extracted from Engelbreth Holm Swarm mouse sarcomas, rich in laminin, collagen, and other ECM proteins. Please take this example, Matrigel has been used for more than four decades for a myriad of cell cultures, (and due to its murine origin is tumorigenic, making it impossible to be used in humans) to be critical on other sections of the manuscript and suggest alternative solutions, not only describe the techniques/scientific tools. The authors did touch on these issues in the Future Outlook section, but still, I believe authors should expand the manuscript with their criticism. Please add this to the different sections of the manuscript. I believe this would increase the overall quality of the manuscript, for sure it will be a plus.

Sections describing and defining Organoids, Organs-On-A-Chip, and D Brioprinting are extensive, well documented, well written, and supported with clear and elucidative Figures and Tables. These elements are well-mentioned and integrated into the presented manuscript. Legends are precise and are in the correct position in the text.  Please format the legends of all figures to be sized as the Figures. Do the same work for the Tables.

The manuscript refers to important guidelines of clinical trial regulation such as “FDA Modernization Act 2.0” which is important to be known in the scientific field. This fact is a positive signal of the author’s care and attention that is appreciated.

In “Future Outlook” section is mentioned by the authors the advantage and disadvantages of certain techniques, that has to be considered a good point. Authors propose, still in this section, possible alternative routes to envision solutions to the actual limitations/disadvantages’ in the field aiming to mimic in vitro physiological issues.

In an overall view, the presented manuscript is, again, well-structured and written, supported by suitable figures and tables that resemble the scientific panorama of part of “Preclinical testing Techniques”. Bibliographic support is correct and very detailed. 

I do believe that this manuscript is able to be published after minor considerations as those written before.

Review of the manuscript: Preclinical Testing Techniques: Paving the Way to New Oncology Screening Approaches

In an overall view, the presented manuscript is, again, well-structured and written, supported by suitable figures and tables that resemble the scientific panorama of part of “Preclinical testing Techniques”. Bibliographic support is correct and very detailed. 

I do believe that this manuscript is able to be published after minor considerations as those written before.

Author Response

Dear Reviewer 2,

First of all, we would like to thank the reviewers for their comments and feedback. Our manuscript titled “Preclinical Testing Techniques: Paving the Way to New Oncology Screening Approaches” has been significantly improved according to the feedback of the reviewers. We have tried to address most of the comments of the reviewers and the details can be found below. The reviewer questions can be found in bold, our answers can be found in italic, and the changes done in the manuscript can be found in blue.

The manuscript was updated, and the changes can be observed by the use of track changes. We hope that the changes created in the manuscript are satisfactory and we welcome any additional feedback.

Dr. Karolina Valente

Reviewer #2:

  • Simple summary and abstract are well-structured and written. The reading is clear and the main points are easily understandable. In the Abstract, line 26, please change the word “realistic” to another one. Realistic doesn’t seem to be the best term in the context of the sentence.

The word has been changed to “physiologically relevant”.

“3D tissue cultures address many of the shortcomings of 2D cultures by more closely replicating the tumour microenvironment through a combination of physiologically relevant drug diffusion, paracrine signaling, cellular phenotype, and vascularization that can better mimic native human tissue.”

  • In the introduction part, at line 46 and 47, please take not that cells are subjected to drugs not the other way around. Studies are done to verify the cellular outcome of certain drugs. Please change the orientation of this sentence.  

This has been modified in the text.

“Additionally, drugs can block specific signaling pathways in target cells, which can then be lysed and analyzed through biochemical or multi-omic approaches [2].”

  • At line 60, please re-insert reference 3. This appears as a superscript and not with [ ]. Please check and correct this with your Mendeley/Endnote bibliographic editor software. Synchronize all the references

Thank you. This has been modified in the text. In addition, all the references have been synchronized.

In addition, animal model results often have weak translatability to their associated clinical trials, and there is less than 8% correlation between in vivo data and clinical trials results [6].

  • In the introduction at line 134, the authors mentioned the fact that 2D cultures do not mimic 3D culture environments or tissue ones. Undeniably, this is absolutely true. Still they should have been more precise on the line 134 “instead of …so they may not predict clinical….” Please state that “they not predict clinical…, cancel the word “may”. The entire scientific community knows and is aware that 2D model don’t mimic at al tissue environments due to many reasons. Please be more critical on the manuscript, if sustained by refs, authors scientific opinion is always welcome.

This sentence has been modified (indicated below). Furthermore, additional sentences have been added to add a critical stance to the manuscript in conjunction with edits from Question 6.

However, 2D cultures do not fully depict human tissue due to the lack of 3D properties, so they do not predict clinical physiological responses to compounds with accuracy. 2D monocultures do not mimic the diverse multicellular environment with multiple cell types and functions found in real tissues. Furthermore, 2D cultures struggle to provide insight on the systemic effects of drugs; 2D cultures can show how effectively a drug can kill cancer cells but cannot reliably predict complex phenomena such as organ toxicity. In addition, in 2D cultures, constant concentration of the drug is deposited across the entire cell monolayer, whereas in vivo tissues will induce a concentration gradient from where the drug is delivered. As a result, drugs can be more effective on cancer cells in 2D, so dosages based on 2D cultures may not be effective in 3D.

  • Methodologies unfolding spheroid formation are quite well described and illustrated in Figure 1. The figure is clear and schematically describes the different methodologies. It should be centered on the page having the text as a reference. Please do the same for Figure legends.  If this concerns to authors, please changed it, otherwise don’t consider this point. Table 1 is well-structured, has scientific references, and is easy to understand.

All figures and figure captions have had their margins adjusted have been centered on the page. All table captions have had their margins adjusted to correspond with the figure captions.

  • Like mentioned before, all sections are well written and designed. Authors made very good effort on the description of laboratory techniques used to perform preclinical tests on oncology broad field. Beyond a description of the current state-of-the-art of the field, I do believe that all the sections lack personal scientific criticism. Authors should explain why e.g. research is done using Matrigel for decades even if spheroids/organoids of different sources, in a regular situation would never be surrounded by a basement-membrane matrix extracted from Engelbreth Holm Swarm mouse sarcomas, rich in laminin, collagen, and other ECM proteins. Please take this example, Matrigel has been used for more than four decades for a myriad of cell cultures, (and due to its murine origin is tumorigenic, making it impossible to be used in humans) to be critical on other sections of the manuscript and suggest alternative solutions, not only describe the techniques/scientific tools. The authors did touch on these issues in the Future Outlook section, but still, I believe authors should expand the manuscript with their criticism. Please add this to the different sections of the manuscript. I believe this would increase the overall quality of the manuscript, for sure it will be a plus.

The following text has been added to the manuscript.

“Organoids are usually cultured in a dome or layer of Matrigel, a soluble basement membrane extract from murine Engelbreth Holm-Swarm sarcoma. Matrigel is composed of laminins, collagen IV, entactin, heparin sulfate proteoglycan perlecan, and a wide variety of growth factors [A]. The concentrations of individual components can vary significantly between batches, especially when considering the growth factors. As a 3D matrix for organoid culture, Matrigel provides an environment that mimics the basement membrane and promotes the growth and differentiation of epithelial cells; however, many cancers such as carcinomas have invaded the adjacent connective tissue and experience an ECM primary consisting of collagen I fibers that is very different than the composition of Matrigel. When culturing organoids derived from invasive carcinomas, researchers should consider the implications that this difference in matrix may have on the results and explore alternative options when feasible.”

  • Sections describing and defining Organoids, Organs-On-A-Chip, and D Brioprinting are extensive, well documented, well written, and supported with clear and elucidative Figures and Tables. These elements are well-mentioned and integrated into the presented Legends are precise and are in the correct position in the text.  Please format the legends of all figures to be sized aras the Figures. Do the same work for the Tables.

This has been modified in the manuscript. The figures have been centered and the captions have been aligned to the left edge of each associated figure.